# The Inhibitory Effect of Plant Extracts on Growth of the Foodborne Pathogen, *Listeria monocytogenes*

**DOI:** 10.3390/antibiotics9060319

**Published:** 2020-06-11

**Authors:** Marina Ceruso, Jason A. Clement, Matthew J. Todd, Fangyuan Zhang, Zuyi Huang, Aniello Anastasio, Tiziana Pepe, Yanhong Liu

**Affiliations:** 1Department of Veterinary Medicine and Animal Production, University of Naples Federico II, 80100 Naples, Italy; marina.ceruso@unina.it (M.C.); aniello.anastasio@unina.it (A.A.); tiziana.pepe@unina.it (T.P.); 2Natural Products Discovery Institute, Baruch S. Blumberg Institute, 3805 Old Easton Rd., Doylestown, PA 18960, USA; jason.clement@bblumberg.org (J.A.C.); matthew.todd@bblumberg.org (M.J.T.); 3Department of Chemical and Biological Engineering, Villanova University, Villanova, PA 19085, USA; fzhang@villanova.edu (F.Z.); zuyi.huang@villanova.edu (Z.H.); 4Eastern Regional Research Center, Agricultural Research Service, U.S. Department of Agriculture, 600 East Mermaid Lane, Wyndmoor, PA 19038, USA

**Keywords:** cell damage, *Listeria monocytogenes*, plant extracts, food safety

## Abstract

*Listeria monocytogenes* is a foodborne pathogen responsible for about 1600 illnesses each year in the United States (US) and about 2500 confirmed invasive human cases in European Union (EU) countries. Several technologies and antimicrobials are applied to control the presence of *L. monocytogenes* in food. Among these, the use of natural antimicrobials is preferred by consumers. This is due to their ability to inhibit the growth of foodborne pathogens but not prompt negative safety concerns. Among natural antimicrobials, plant extracts are used to inactivate *L. monocytogenes*. However, there is a large amount of these types of extracts, and their active compounds remain unexplored. The aim of this study was to evaluate the antibacterial activity against *L. monocytogenes* of about 800 plant extracts derived from plants native to different countries worldwide. The minimal inhibitory concentrations (MICs) were determined, and scanning electron microscopy (SEM) was used to verify how the plant extracts affected *L. monocytogenes* at the microscopic level. Results showed that 12 of the plant extracts had inhibitory activity against *L. monocytogenes*. Future applications of this study could include the use of these plant extracts as new preservatives to reduce the risk of growth of pathogens and contamination in the food industry from *L. monocytogenes*.

## 1. Introduction

Listeriosis is a disease caused by the foodborne pathogen *Listeria monocytogenes*. It is assessed to cause about 1600 illnesses each year in the United States (US), and about 2500 confirmed invasive human cases were reported in European Union (EU) countries [1,2], with a high mortality rate of about 20% in at-risk populations. A wide variety of foods, including meat, dairy products, and fresh produce, are associated with outbreaks of listeriosis. *L. monocytogenes* is able to survive and grow in harsh environmental conditions such as high salt concentration, low pH, and low temperature. This characteristic increases its potential for contamination and growth on food products and poses some challenges for control [3,4,5]. Thus, the presence of *L. monocytogenes* in foodstuffs is still considered a major food safety problem worldwide [6]. Several technologies and antimicrobials are applied to control *L. monocytogenes.* Use of natural antimicrobials is preferred by consumers, particularly when compared with synthetic preservatives. This is due to their ability to inhibit the growth of foodborne pathogens but not prompt negative safety concerns [7]. Among natural antimicrobials, plant extracts are used to inactivate *L. monocytogenes* [8,9,10]. However, there is a large amount of these types of plant extracts, and their active compounds remain largely unexplored. Therefore, there is the need to expand our knowledge about the types and doses of plant extracts useful as antimicrobials for controlling important foodborne pathogens such as *L. monocytogenes*. The aim of this study was to evaluate the antibacterial activity against *L. monocytogenes* strain F2365 of about 800 extracts derived from plants native to different countries from all over the world. Results showed that 12 of the plant extracts had inhibitory activity against *L. monocytogenes*. Scanning electron microscopy (SEM) was used to examine the effects of the extracts on *L. monocytogenes* at the microscopic level. The discovery of new natural antimicrobials could have different applications in the food industry, such as for treatments with new, more effective preservatives/antimicrobials or for the enhancement of those currently used, as well as for the formulation of innovative packaging materials.

## 2. Results

### 2.1. Effect of Plant Extract on Growth of L. monocytogenes and Determination of Minimum Inhibitory Concentrations (MICs)

In total, 780 plant extracts were used to test the growth inhibition in *L. monocytogenes*. These plant extracts are from plants found worldwide and, moreover, different parts (leaves, root, twigs, etc.) of some plants were also included. Twelve plant extracts were shown to inhibit the growth of *L. monocytogenes*. The identity of the plants and the MICs obtained are shown in Table 1.

With regard to the families of the 12 extracts, it is interesting to note that three out of the 12 were derived from different species of the same family of Fabaceae (*Baphia racemosa*, *Desmodium adscendens*, and *Eriosema preptum*), and the rest were derived from different families. Regarding the geographic origin, 50% of the 12 extracts were derived from plants from South Africa, 25% from the Republic of Georgia, and 25% from the US. No hit was obtained from extracts from Puerto Rico, but a very small portion of the tested plant extracts (15 of 780) had this geographic origin. With regard to the MICs, *Baphia racemosa* and *Sansevieria hyacinthoides* showed the lowest value of MICs (2.5 mg/mL) followed by *Passiflora foetida*, *Desmodium adscendens*, *Salvia nemorosa*, *Alnus barbata*, and *Botrychium multifidum* at the MIC values of 5 mg/mL. The species *Trichilia emetica*, *Eriosema preptum*, *Darlingtonia californica*, *Proboscidea louisianica*, and *Sambucus ebulus* had MIC values of 10 mg/mL.

### 2.2. Scanning Electron Microscopy (SEM)

The morphology of *L. monocytogenes* F2365 cells treated with the 12 plant extracts (10 mg/mL) was observed using SEM. The untreated *L. monocytogenes* cells showed a normal cell morphology, with a typical short rod shape, intact cell structure, a smooth and compact surface, and flagella (Figure 1C, Figure 2C and Figure 3C). After treatment at 30 °C for 24 h, *L. monocytogenes* cells showed loss of flagella in all of the samples. Treatment with seven out of the 12 plant extracts showed consistently damaged *L. monocytogenes* cells (Figure 1, Figure 2 and Figure 3). In particular, the cells showed morphological damage such as detachment of the cytoplasmic membrane from the cell wall, leakage of intracellular components, and severe cell collapse and deformation. The effect of No. 2 plant extract (*Passiflora foetida*) was peculiar. Only for this extract, there was the creation of holes in the external wall of the bacteria. Extract No. 12 (*Botrychium multifidum*) had a very strong effect on *L. monocytogenes* cells that were practically destroyed. Meanwhile, the cells of *L. monocytogenes* treated with extract No. 1, 4, 5, 7, and 8 shown in Figure 1, Figure 2 and Figure 3 displayed minor damage. Table 2 reports a summary of the SEM results.

## 3. Discussion

In this paper, the potential antibacterial activity of an extraordinarily vast collection of plant extracts was explored against the foodborne pathogen *L. monocytogenes.* This collection included extracts from plants found all over the world and including different parts of the plant. Furthermore, the extracts were separated by different solvents based on polarity (polar: methanol, dichloromethane; non-polar: hexane), which makes further fractionation/separation much easier.

This research provides an expansion of knowledge on the antibacterial properties of plant extracts and identified 12 new anti-*Listeria* plant extracts. None of these extracts, to our knowledge, were tested before for anti-listerial behavior. Considering the activities identified in previous research, we can separate our 12 plant extracts as belonging to three groups of plants: A—plants already used with medical application, showing antimicrobial activities against other bacteria; B—plants with other medical functions but not antibacterial; C—plants that previously never showed any antimicrobial activities or medical applications.

The A group includes *Trichilia emetica, Passiflora foetida, Salvia nemorosa*, and *Sambucus ebulus*. *T. emetica* (plant extract No. 1) previously showed hepatoprotective and antibacterial activity, including action against *Staphylococcus aureus* [11], and it was also involved in malaria control [12]. *P. foetida* is a wild species commonly used for the treatment of hysteria, asthma, skin diseases with inflammation [13], and for many other anti-inflammatory and analgesic activities [14]. It also showed an antibacterial effect, e.g., against *Escherichia coli* HB 101, *Streptococcus pyogenes*, and *Bacillus thuringiensis* strains [15,16]. *S. nemorosa* deserves special mention since it gave very good results against *L. monocytogenes* in this study (Table 2). Several species of the genus *Salvia* are known for their uses as additives in food products [17,18,19]. The extract obtained from the species *S. nemorosa* was never previously tested against *L*. *monocytogenes* but appeared to have good antibacterial activity against other foodborne pathogens [20]. The last plant of this group is *Sambucus ebulus* (plant extract No. 11), and it has several clinical applications [21,22], with anti-inflammatory components of the leaves [23] and antioxidant activity of the flowers [24,25]. Its methanolic extract had antimicrobial activity against *S. aureus* [26,27]. 

Plants belonging to the B group are *Baphia racemose, Sansevieria hyacinthoides*, *Desmodium adscendens,* and *Eriosema preptum. B. racemose* (our plant extract No. 2) showed in previous research a specific inhibition effect on human liver β-d-glucuronidase and α-l-iduronidase [28]. *S. hyacinthoides* (plant extract No. 3) is one of the most promising extracts against *L. monocytogenes* identified in this study. Most members of *Sansevieria* are of great economic importance as ornamentals [29], but *S. hyacinthoides* is also used in traditional African medicine. Leaves and rhizomes of the plant are squeezed, and the juice is used in the treatment of ear infections, toothaches, ulcers, intestinal worms, stomach disorders, and diarrhea [30]. It also can be used to treat snake bites [31,32,33]. However, no use is made of this plant, as far as we know, in food hygiene, and our results show that its use warrants further investigation. In the current research, *S. hyacinthoides* showed good results for *L. monocytogenes* growth inhibition and a low MIC of 2.5 mg/mL, and it induced damage to bacterial cells observed with SEM (Figure 2). The *B. racemose* and *S. hyacinthoides* extracts gave the same results in regard to *L. monocytogenes* growth inhibition and MIC, but the effect on the bacterial cells observed using SEM appeared less obvious for the *B. racemose* extract, since only flagella were lost. Regarding plant extract No. 5, an ethanol extraction of *D. adscendens* had antipsychotic-like activities in mice [34]. *E. preptum* is the plant species of No. 6 extract. Several *Eriosema* (Fabaceae) species are a good source of flavonoids with pharmacological activities [35,36], but no anti-bacterial activities were reported for the *E. preptum* species. 

Finally, the C group comprises plants that, to our knowledge, did not previously demonstrate antimicrobial activities or medical applications; however, antibacterial potential was observed for the first time in in this study through their good anti-listerial activity. The species with these features were *Darlingtonia californica*, *Proboscidea louisianica*, *Alnus barbata,* and *Botrychium multifidum. D. californica* belongs to Sarraceniaceae, a carnivorous plant family, endemic to southern Oregon and northern California, United States of America (USA) [37]. Devil’s claw (*P. louisianica*) is basically grown as an ornamental plant all over the world (USA, Africa, Australia, Europe) and is a source of essential oil [38]. *A. glutinosa* subsp. *barbata* is found in various countries, including Turkey and Iran, and it is common to the Colchis forests of Georgia [39]. There are no reported uses of this taxon. *B. multifidum* is a fern native to California, also found elsewhere in North America and beyond, with no particular use or trades. This group of plant extracts warrants further investigation.

With regard to the mechanism of action of the plant extracts, it is known that plant polyphenols play important roles in plant defense mechanisms against bacteria, viruses, and fungi, and that they are the major components of plant extracts [40,41]. Plants belonging to group A in the current study were reported to have antibacterial activity mainly due to the presence of polyphenols [11,16,18,26,42], and their use was described as potential natural preservatives for the food industry [40,41,43,44,45]. The mechanism of bacterial inhibition by polyphenols is complex. They can act by chelating iron, which is important for the survival of many bacteria [46]. They also work together with nonspecific forces, producing an effect on microbial membranes, adhesins, enzymes, and cell envelope transport proteins, and polyphenols interact with proteins and/or phospholipids from the lipid bilayer, increasing membrane permeability, modifying ion transport processes, and damaging cell membranes [47,48,49,50,51,52,53,54]. Polyphenols may also induce irreversible changes in *E. coli*, *Pseudomonas aeruginosa*, *S. aureus*, and *L. monocytogenes* membranes, causing rupture and pore formation with leakage of intracellular constituents [55]. Furthermore, polyphenols inhibit bacterial motility through loss of flagella [41,56], and they can inhibit biofilm formation [41,57]. For example, polyphenols from olive leaves and olive mill wastes were shown to reduce expression of motility- and biofilm-related genes in *E. coli* [58]. 

Food-related applications for these plant extracts could be for the formulation of new food preservatives and sanitizers for the food processing environment, as well as for the development of innovative food packaging. The MIC values in the mg/mL (g/L) range are typical for plant extracts since they contain a mixture of active and non-active compounds. For example, olive leaf extract that has antimicrobial activities had a MIC of 62.5 mg/mL for *L. monocytogenes* [57,59]. In terms of application in food, the extracts or their active compounds can be used in combination with other antimicrobials to achieve synergistic effects. Moreover, their MICs could be further reduced if they are used as nanowires [59].

In the current study, inhibition of *L. monocytogenes* seemed compatible with the described effects caused by plant polyphenols, since there was inhibition of bacterial growth, cell membrane damage, and pore formation with some extracts, and, in all cases, there was loss of flagella (Table 2). Future directions will focus on fractionation of the 12 plant extracts to reveal their active compounds. No toxicities of these plants are yet reported. Once the active compounds in the extracts are identified, their toxicities should be determined using mouse models to define the safe dose. In addition, the cytotoxicity, mutagenicity, and genotoxicity of the plant extracts should also be investigated. 

In addition, gene expression analyses of *L. monocytogenes* exposed to these plant extracts may reveal their potential effects on virulence-related genes of *L. monocytogenes* to determine if there could be a decrease in virulence of *L. monocytogenes* with exposure to the extracts or the active compounds. 

## 4. Materials and Methods 

### 4.1. Plant Extracts and Crude Fractionation

All of the plant families tested in this study are shown in Figure 4.

The plant extracts tested in this study were supplied and identified by the Natural Products Discovery Institute, Baruch S. Blumberg Institute, Doylestown, Pennsylvania, USA. The extracts were prepared from specimens of *n* = 780 plant species representing *n* = 326 plant species from the Republic of South Africa, *n* = 285 from the US, *n* = 154 plant species from the Republic of Georgia, and *n* = 15 from Puerto Rico (Figure 5). 

The extraction was performed according to the following procedure: the plant material (either whole plants or separated parts) for each extract was dried and ground up into a powder. Methanol was added to each powder sample (12 mL/g), and this was shaken for about one hour. The solvent mixture was filtered, and the filtrate was concentrated under vacuum. The concentrated extracts were transferred to amber Qorpak bottles and stored at −20 °C. The crude methanol extracts (about 2 g of each) were suspended in 20 mL of 90% aqueous methanol and washed twice with an equal volume of hexane. The hexane washes were pooled and concentrated under vacuum to yield the hexane fractions. The aqueous methanol layer was concentrated and resuspended in 20 mL of 50% aqueous methanol, and this was washed twice with an equal volume of dichloromethane. The dichloromethane washes were pooled and dried under vacuum to afford the dichloromethane fractions. The 50% aqueous methanol fraction was added at about 5 cc of pre-wet solid polyvinylpyrrolidone (PVP), and this was shaken for 2 h. After centrifugation and collection of the 50% aqueous methanol solution, the PVP was washed with 20 mL of methanol for 2 h. The samples were again centrifuged, the methanol solutions were collected, and the two PVP washes were pooled and dried for each crude extract to afford the detanninized aqueous methanol fraction. For screening, solutions of the hexane, dichloromethane, and detanninized aqueous methanol fractions were prepared at 10 mg/mL in dimethyl sulfoxide (DMSO), which were later diluted for assay. 

### 4.2. L. monocytogenes Strain and Culture Conditions

*L. monocytogenes* F2365 was used in this study; it was isolated from Mexican-style soft cheese that was implicated in an outbreak of listeriosis in the US [60]. The strain was stored in brain heart infusion (BHI) broth with 20% glycerol (*v*/*v*) at −80 °C. When used in experiments, it was streaked onto BHI agar plates and incubated at 37 °C for 24 h. A typical colony was selected and inoculated into 5 mL of BHI and incubated at 37 °C with shaking at 150 rpm for 24 h.

### 4.3. Plant Extracts Effect on L. monocytogenes Growth and Determination of MICs

For examination of the effect of the plant extracts on *L. monocytogenes*, a 1:1000 dilution of the overnight culture was made with fresh BHI broth. A 96-well plate was used for testing. The wells in the first column received 190 µL of BHI broth and 10 µL of DMSO (negative control). Wells in the last column of the plate were used as the positive control receiving 10 µL of DMSO (0.1 mg/mL) and 190 µL of the diluted *L. monocytogenes* culture. The central wells received 10 µL of the plant extract stock solution tested and 190 µL of the diluted *L. monocytogenes* culture. The plate was placed into a microplate reader Epoch2 (BioTek, Winooski, VT, USA) and incubated at 37 °C for 24 h, with optical density at 600 nm (OD_600_) readings recorded once every hour. For each plant extract, a minimum of three replicates were performed. 

The MICs for *L. monocytogenes* F2365 were determined using a two-fold dilution method [61] for the plant extracts that showed growth inhibition in the screening assay. Bacterial cells were treated with different concentrations of the plant extracts (10, 5, 2.5, 1.25, 0.625, 0.3125, 0.156, and 0.078 mg/mL) in a 96-well plate. The wells in the first column received 190 µL of BHI broth (negative control) and 0.1 mg/mL of DMSO. The final column of the plate was used as a positive control receiving 200 µL of the diluted *L. monocytogenes* culture. The central wells received 10 µL of the plant extract solution to test in the top row plate, two-fold diluted moving down from the top, and 190 µL of the diluted *L. monocytogenes* culture. The plate was placed into a microplate reader Epoch2 (BioTek, VT, USA), incubated at 37 °C for 24 h, and OD_600_ readings were recorded once every hour. The lowest extract concentrations that showed no increase of OD_600_ over 24 h were determined as the MIC. Three independent experiments were performed to assess MICs. 

### 4.4. Scanning Electron Microscopy

Scanning electron microscopy was used to determine how the plant extracts affected the bacteria at the microscopic level. We were able to observe the surface of the bacteria, as well as flagella production. A 5-mL culture of *L. monocytogenes* F2365 was supplemented with the plant extracts at MICs. The culture was incubated overnight at 30 °C at 200 rpm. One hundred microliters of the culture was pipetted onto a 12-mm microcover glass slide (Thermo Scientific Portsmouth NH, USA) and allowed to adhere for 10 min. After 30 min, 500 µL of 2.5% glutaraldehyde was added to the slide, covering the area with the bacteria, and fixation was for 30 min. This was followed by a 30-min wash with 2–3 mL of the following solutions: 0.1 M imidazole (two consecutive washes), 50% ethanol, 80% ethanol, and 90% ethanol, followed by three consecutive washes with 100% ethanol. Samples were stacked into a wire basket, separated by a cloth, and placed into a critical point drying apparatus (Denton DCP-1) that used liquid carbon dioxide to completely dry the sample. The samples were removed from the critical dryer and mounted onto stubs. They were then sputter gold-coated for 1 min. Samples were finally viewed with the FEI Quanta 200 F Scanning Electron Microscope (FEI Co., INC., Hillsboro, OR, USA), with an accelerating voltage of 10 kV in high vacuum mode. Instrumental magnification was set at 10,000×, 50,000×, and 100,000× for imaging purposes.

## 5. Conclusions

This study showed the anti-*L. monocytogenes* activity of 12 extracts derived from plants originating from different countries. Results from our investigation demonstrate that these natural extracts notably inhibited *L. monocytogenes* growth. SEM results demonstrated that some plant extracts had a disruptive effect on the cell membrane, causing serious damage of the membrane of *L. monocytogenes,* as well as causing loss of flagella. Further fractionation of these extracts will provide detailed identification of the active components, with potential interesting applications as food preservatives to reduce the risk of contamination in the food industry from *L. monocytogenes*.

## Figures and Tables

**Figure 1 antibiotics-09-00319-f001:**
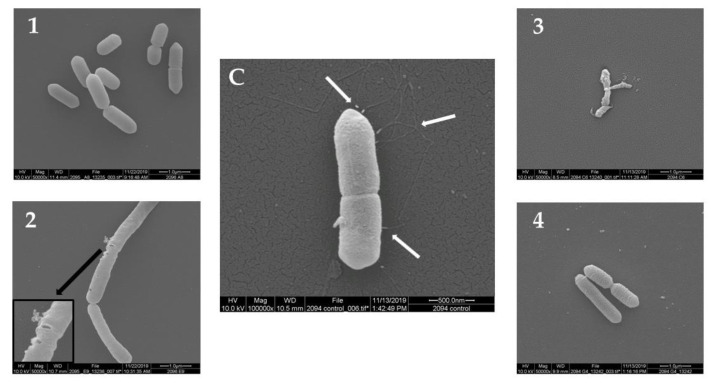
Morphology of *L. monocytogenes* F2365 cells treated with plant extracts No. (**1–4**) (group A) observed using SEM, in comparison with *L. monocytogenes* F2365 that was not treated with samples—control (**C**). White arrows show flagella in (**C**). The black arrow shows pores in the cell membrane. Number/plant species correspondence is shown in Table 1.

**Figure 2 antibiotics-09-00319-f002:**
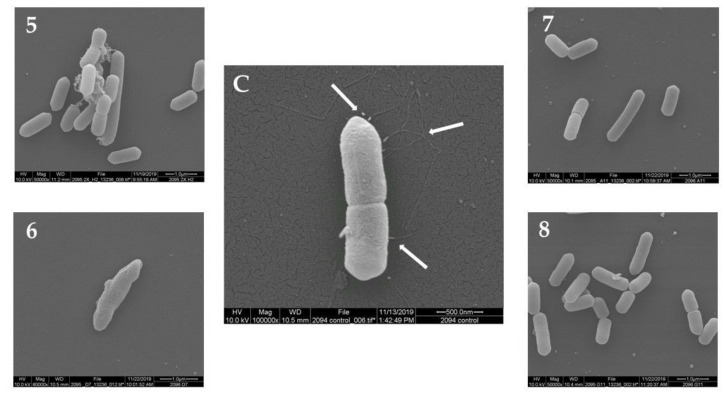
Morphology of *L. monocytogenes* F2365 cells treated with the plant extracts No. (**5–8**) (group B) observed using SEM, in comparison with *L. monocytogenes* F2365 that was not treated with samples—control (**C**). White arrows show flagella in (**C**).

**Figure 3 antibiotics-09-00319-f003:**
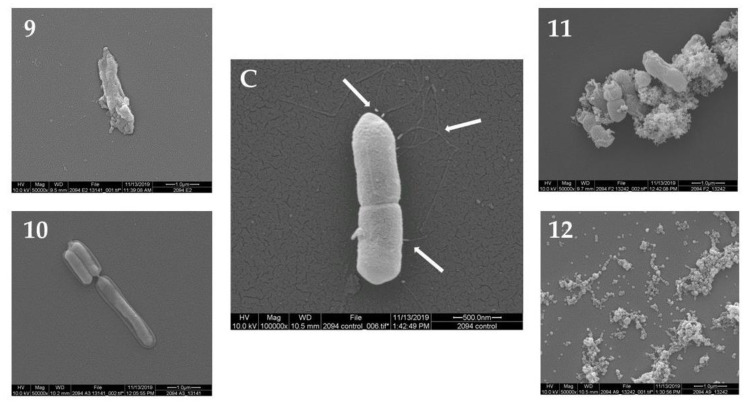
Morphology of *L. monocytogenes* F2365 cells treated with the plant extracts No. (**9–12**) (group C) observed using SEM, in comparison with *L. monocytogenes* F2365 that was not treated with samples—control (**C**). White arrows show flagella in (**C**).

**Figure 4 antibiotics-09-00319-f004:**
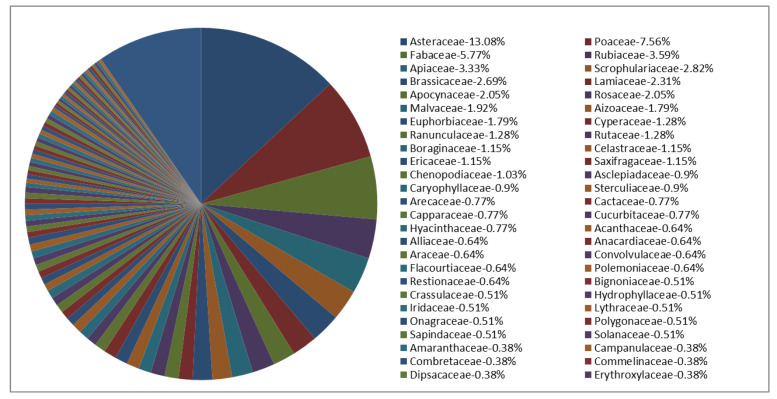
Families of the plants and respective percentage of use in this study.

**Figure 5 antibiotics-09-00319-f005:**
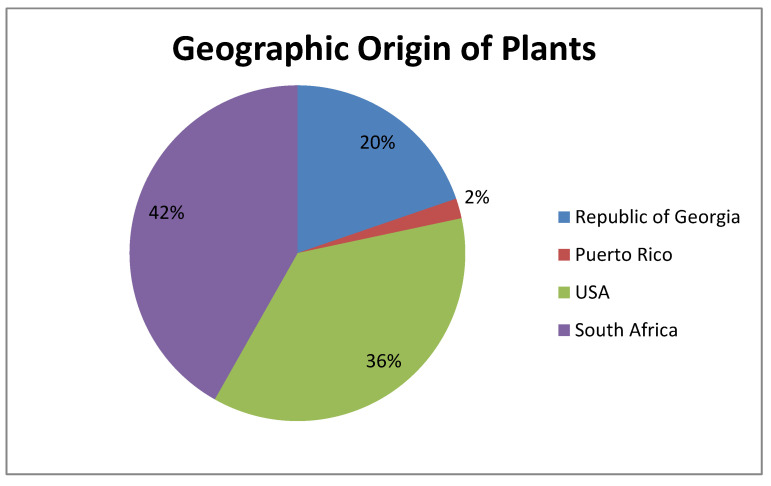
Geographic origin of the plants and respective percentage of use in this study.

**Table 1 antibiotics-09-00319-t001:** The 12 plant extracts with antibacterial effect against *Listeria monocytogenes* F2365. MIC—minimum inhibitory concentration; USA—United States of America.

Group	No.	Family	Genus	Species	Extraction Origin (Plant Part)	Geographicorigin	MIC (mg/mL)
A	1	Meliaceae	*Trichilia*	*emetica*	Leaves	South Africa	10
	2	Passifloraceae	*Passiflora*	*foetida*	Whole plant	South Africa	5
	3	Lamiaceae	*Salvia*	*nemorosa*	Whole plant	Georgia	5
	4	Sambucaceae	*Sambucus*	*ebulus*	Whole plant	Georgia	10
B	5	Fabaceae	*Baphia*	*racemosa*	Root	South Africa	2.5
	6	Dracaenaceae	*Sansevieria*	*hyacinthoides*	Root	South Africa	2.5
	7	Fabaceae	*Desmodium*	*adscendens*	Whole plant	South Africa	5
	8	Fabaceae	*Eriosema*	*preptum*	Whole plant	South Africa	10
C	9	Sarraceniaceae	*Darlingtonia*	*californica*	Leaves	USA	10
	10	Pedaliaceae	*Proboscidea*	*louisianica*	Seed pod	USA	10
	11	Betulaceae	*Alnus*	*barbata*	Leaves + twigs	Georgia	5
	12	Ophioglossaceae	*Botrychium*	*multifidum*	Root	USA	5

**Table 2 antibiotics-09-00319-t002:** Summary of the results obtained by SEM observations of the bacterial cells.

Group	No.	Genus	Species	MIC (mg/mL)	Cell Damage (SEM)
A	1	*Trichilia*	*emetica*	10	Loss of flagella
	2	*Passiflora*	*foetida*	5	Loss of flagella, holes in the external bacteria wall
	3	*Salvia*	*nemorosa*	5	Loss of flagella, severe cell collapse and deformation
	4	*Sambucus*	*ebulus*	10	Loss of flagella
B	5	*Baphia*	*racemosa*	2.5	Loss of flagella
	6	*Sansevieria*	*hyacinthoides*	2.5	Loss of flagella, cell deformation
	7	*Desmodium*	*adscendens*	5	Loss of flagella
	8	*Eriosema*	*preptum*	10	Loss of flagella
C	9	*Darlingtonia*	*californica*	10	Loss of flagella, severe cell collapse and deformation
	10	*Proboscidea*	*louisianica*	10	Loss of flagella, leakage of intracellular components
	11	*Alnus*	*barbata*	5	Loss of flagella, severe cell collapse and deformation
	12	*Botrychium*	*multifidum*	5	Cell destruction

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
