# Peer review of "The Inhibitory Effect of Plant Extracts on Growth of the Foodborne Pathogen, Listeria monocytogenes"

_antibiotics, 2020, doi:10.3390/antibiotics9060319_

Round 1

Reviewer 1 Report

The paper gives useful information about a wide range of different plant extracts, and the data presented can be used as a basis for further specific investigations.

Some improvements should be made, as described below.

Abstract

  • Line 18 (and introduction, line 36): about 2500 cases (refer to the last report)
  • Line 30: it is the risk of contamination or of growth on foodstuffs? (which are the potential applications? See below)

Keywords

  • Line 32: some are not specific (avoid “food control”); “cell damage” could be used, to highlight the use of electron microscopy

Introduction

It is very short; also if extended information about the plant extracts is given in the discussion, a brief presentation of the main functions that can be exerted by plant extracts in the food industry should be added.

  • Line 41: potential for contaminating but also for growth on foodstuffs
  • Lines 43-44: the concept of this proposition (from “however” to “control”) is already given at line 39; these parts should be merged

Results

  • Lines 59-60: repetition of the number ((line 60: these plant extracts)
  • Figure captions (lines 91, 96, 102): Just “morpfhology” (not “cell morphology”)
  • Figure captions (lines 92, 97, 103): “non treated samples”
  • Line 107: “Summary of the results obtained by SEM observations of the bacterial cells”
  • Table 2, 8th row: Loss (not losts)

Discussion

Is there any information about toxicity of the plants used? This could be an important information for the application in food production.

Some more information about the potential use of these compounds could be useful for the reader (e.g. examples of possible application: typologies of foodstuffs, production or packaging phase?)

  • Line 118: belonging (not appertaining)
  • Lines 123-124: “including an action against”
  • Line 127: “E. coli HB 101, Streptococcus…”
  • Line 133: anti-inflammatory components
  • Line 133: Its methanolic extract..
  • Line 138: No. (capital)
  • Line 142: also is repeated two times
  • Line 164: play important roles
  •  

Materials and methods

  • Line 191, Caption of figure 4: avoid the repetition of “in this study”
  • Lines 194-195: the term “families” should be changed to “species”?
  • Lines 200-201: “Methanol was added to each…”
  • Line 201: about one hour
  • Line 249: fixation lasted for

References

Some of the references seem not appropriated, as general information is taken from the introduction of very specific studies, or the item seems different from the information taken (references No 3, 4, 5, 7, 8, 13, 67).

Check the format of the references, and use the same format for all the references (e.g. “and” for the last author should not be used; citations of books with different formats)

  • Line 286-288: The EFSA reference should be updated
  • Line 312: Williams, L.
  • Line 384. Dot after Thunb
  • Check if the names of bacteria are written in Italics
  • Lines 477-478: put the spaces before and after Escherichia coli.

Author Response

09 June 2020

Dear Editor,

We have addressed the reviewers’ comments.  Please see answers to their comments below and changes made in the manuscript. 

We thank the reviewers for their comments, and we hope that the manuscript is now acceptable for publication.

Sincerely,

Yanhong Liu, Ph.D.

Reviewer #1:

Abstract

Line 18 (and introduction, line 36): about 2500 cases (refer to the last report)

The number of cases  was updated to 2500 on line 18 and line 36.

Line 30: it is the risk of contamination or of growth on foodstuffs? (which are the potential applications? See below)

The sentence has been corrected with “The risk of growth and contamination”.

Keywords

Line 32: some are not specific (avoid “food control”); “cell damage” could be used, to highlight the use of electron microscopy

The keyword “food control” was replaced with “cell damage”.

Introduction

It is very short; also if extended information about the plant extracts is given in the discussion, a brief presentation of the main functions that can be exerted by plant extracts in the food industry should be added.

The following sentence was added “The discovery of new natural antimicrobials could have different applications in the food industry, such as for treatments with new, more effective preservatives/antimicrobials or for the enhancement of those currently used, as well as for the formulation of innovative packaging materials”.

Line 41: potential for contaminating but also for growth on foodstuffs

The sentence has been corrected to “potential for contamination and growth”.

Lines 43-44: the concept of this proposition (from “however” to “control”) is already given at line 39; these parts should be merged

The sentences were merged.

Results

Lines 59-60: repetition of the number ((line 60: these plant extracts)

The number repetition was deleted             

Figure captions (lines 91, 96, 102): Just “morpfhology” (not “cell morphology”)

“cell morpphology” was replaced by “morphology” in the figure captions.

Figure captions (lines 92, 97, 103): “non treated samples”

Figure captions were corrected.

Line 107: “Summary of the results obtained by SEM observations of the bacterial cells”

 “Summary of the results obtained by SEM observations of the bacterial cells” was added as the title of table2

Table 2, 8th row: Loss (not losts)

Done

Discussion

Is there any information about toxicity of the plants used? This could be an important information for the application in food production.

We searched the literature and no toxicities of these plants have been reported yet.

Thus, the following sentence was added to discussion: “No toxicities of these plants have been reported yet. Once the active compound in the extracts are identified, their toxicities should be determined using mouse models to define the safe dose.  In addition, the cytotoxicity, mutagenicity and genotoxicity of these plant extracts should also be investigated“.      

Some more information about the potential use of these compounds could be useful for the reader (e.g. examples of possible application: typologies of foodstuffs, production or packaging phase?)

The following sentence was added to discussion: “Food-related applications for these plant extracts could be for the formulation of new food preservatives and sanitizers for the food processing environment, and for development of innovative food packaging”.                                                                                                                                              

Line 118: belonging (not appertaining)

Done

Lines 123-124: “including an action against”

Done

Line 127: “E. coli HB 101, Streptococcus…”

Done

Line 133: anti-inflammatory components

Done

Line 133: Its methanolic extract..

Done

Line 138: No. (capital)

Done

Line 142: also is repeated two times

One “also” was deleted

Line 164: play important roles

Done

Materials and Methods

Line 191, Caption of figure 4: avoid the repetition of “in this study”

The repetition was deleted

Lines 194-195: the term “families” should be changed to “species”?

Yes, they were replaced

Lines 200-201: “Methanol was added to each…”

Done

Line 201: about one hour

Done

Line 249: fixation lasted for

Done

References          

Some of the references seem not appropriated, as general information is taken from the introduction of very specific studies, or the item seems different from the information taken (references No 3, 4, 5, 7, 8, 13, 67).

Done. References 3, 4, 5, 7, 8, 13, 67 were taken out from the manuscript.

Check the format of the references, and use the same format for all the references (e.g. “and” for the last author should not be used; citations of books with different formats)

The reference have been re-formated.

Line 286-288: The EFSA reference should be updated

The EFSA reference was updated

Line 312: Williams, L.

Done

Line 384. Dot after Thunb

Done

Check if the names of bacteria are written in Italics

Done

Lines 477-478: put the spaces before and after Escherichia coli.

Done

Reviewer 2 Report

Marina Ceruso et al demonstrate the antimicrobial effect of several plant extracts of various origin in this interesting manuscript. The manuscript is well written, with sound English, the results are clearly presented and contain enough novelty to be published in Antibiotics, however only after minor revisions.

My concerns:

The MIC values presented in the manuscript are quite high in the mg/ml (g/L) range. I think the practical applicability of these extracts is therefore limited, because they may significantly alter the properties of the original foodstuff. The authors are suggested to address this problem in the manuscript.

The MIC values presented are determined for Partial or Total criteria (MICPI or MICTI)? What was the error of the results? Did the values change?

The decimal separator should be corrected to point (.) instead of comma (,). Please check Figure 4.

The space between the number and unit is missing in some places, for example line 78, 203, 219-220, 229, 238…

Author Response

09 June 2020

Dear Editor,

We have addressed the reviewers’ comments.  Please see answers to their comments below and changes made in the manuscript. 

We thank the reviewers for their comments, and we hope that the manuscript is now acceptable for publication.

Sincerely,

Yanhong Liu, Ph.D.

Reviewer #2:

Marina Ceruso et al demonstrate the antimicrobial effect of several plant extracts of various origin in this interesting manuscript. The manuscript is well written, with sound English, the results are clearly presented and contain enough novelty to be published in Antibiotics, however only after minor revisions.

My concerns:

The MIC values presented in the manuscript are quite high in the mg/ml (g/L) range. I think the practical applicability of these extracts is therefore limited, because they may significantly alter the properties of the original foodstuff. The authors are suggested to address this problem in the manuscript.

“The MIC values in the mg/ml (g/L) range are typical for plant extracts since they contain a mixture of active and non-active compounds.  For example, olive leaf extract that has antimicrobial activities had a MIC of 62.5 mg/ml for L. monocytogenes (Liu et al, 2017, Du et al., 2020).  In terms of application in food, the extracts or their active compounds can be used in combination with other antimicrobials to achieve synergystic effects.  Also, their MICs could be futher reduced if they are used as nanowires (Du et al., 2020)”.    These sentences were added to Discussion section.

The MIC values presented are determined for Partial or Total criteria (MICPI or MICTI)? What was the error of the results? Did the values change?

We used a 2-fold dilution method to perform the MIC test (Balouiri  et al., 2016).   The lowest extract concentrations that showed no increase of OD600 over 24 h was determined as the MIC.  The concentrations for inhibition were the same after 3 independent experiments. Therefore, they were defined as the MICs.

The decimal separator should be corrected to point (.) instead of comma (,). Please check Figure 4.

Corrected.

The space between the number and unit is missing in some places, for example line 78, 203, 219-220, 229, 238…

All the spaces between number and unit were added.

Reference

Balouiri, M., Sadiki, M., and Ibnsouda, S. K. (2016). Methods for in vitro evaluating antimicrobial activity: a review. J. Pharm. Anal. V 6, 71–79. doi: 10.1016/j.jpha.2015.11.005

Du R , Qu Y , Qi PX , Sun X , Liu Y , Zhao M . 2020. Natural flagella-templated Au nanowires as a novel adjuvant against Listeria monocytogenes. Nanoscale. 12(9):5627-5635. doi: 10.1039/c9nr10095d.
